# The Change in Growth, Osmolyte Production and Antioxidant Enzymes Activity Explains the Cadmium Tolerance in Four Tree Species at the Saplings Stage

Zikria Zafar [1,2], Fahad Rasheed [1,*], Waseem Razzaq Khan [3,4], Muhammad Mohsin [5], Muhammad Zahid Rashid [6], Mohamad Maulana Magiman [7], Zohaib Raza [1], Zamri Rosli [4], Shazia Afzal [8] and Fauziah Abu Bakar [9,*]

1 Department of Forestry and Range Management, University of Agriculture, Faisalabad 38040, Pakistan
2 Department of Forest Genetics and Forest Tree Breeding, University of Goettingen, Busgenweg 2, 37077 Goettingen, Germany
3 Institut Ekosains Borneo, Universiti Putra Malaysia, Kampus Bintulu, Bintulu 97008, Malaysia
4 Department of Forestry Science, Faculty of Agricultural and Forestry Sciences, Universiti Putra Malaysia, Kampus Bintulu, Bintulu 97008, Malaysia
5 School of Forest Sciences, University of Eastern Finland, 80100 Joensuu, Finland
6 Horticultural Research Institute, Ayub Agriculture Research Institute, Faisalabad 38950, Pakistan
7 Faculty of Humanities, Management and Science, Universiti Putra Malaysia, Kampus Bintulu, Bintulu 97008, Malaysia
8 Department of Forestry, College of Agriculture, University of Sargodha, Sargodha 40100, Pakistan
9 Department of Crop Science, Faculty of Agricultural and Forestry Sciences, Universiti Putra Malaysia, Kampus Bintulu, Bintulu 97008, Malaysia
* Correspondence: fahad.rasheed@uaf.edu.pk (F.R.); ab_fauziah@upm.edu.my (F.A.B.); Tel.: +92-333-0689727 (F.R.); +60-86-855839 (F.A.B.)

**Abstract:** Phytoremediation is a green technology; however, very few species of arid environments have been identified as hyperaccumulators and fast growers. Therefore, a greenhouse experiment was performed to evidence the phytoaccumulation potential of *Conocarpus erectus*, *Syzygium cumini*, *Populus deltoides* and *Morus alba* at the sapling stage. Six-month-old plant saplings were subjected to control (CK; 0 μM) and cadmium treatments (Cd; CdCl$_2$; 200 μM). The results depicted that plant growth, dry biomass production (leaf and stem) and chl *a*, *b* and carotenoid contents decreased significantly in all four species under Cd treatment; however, the lowest decrease was evidenced in *Conocarpus erectus*. The concentration of hydrogen peroxide and superoxide radical increased significantly in all four species, with the highest increase observed in *Morus alba*. Osmolytes production, antioxidant enzymes activity (superoxide dismutase, peroxidase, catalase and ascorbate peroxidase) and Cd accumulation in the leaves, stem and root increased significantly in all four species under Cd treatment, with the highest increase observed in *Conocarpus erectus*. The translocation factor was >1 in *Conocarpus erectus*, *Syzyngoim cumini* and *Populus deltoides* and was <1 in *Morus alba*. The study revealed a better Cd tolerance in Conocarpus erectus, which was driven by the effective osmolyte balance and antioxidant enzymes mechanism.

**Keywords:** dry biomass; phytoaccumulation; *Conocarpus erectus*; *Syzyngium cumini*; *Populus deltoides*; *Morus alba*; oxidants; antioxidant

## 1. Introduction

Globally, heavy metals (HMs) pollution of the natural environment has become a serious concern. The HMs, which are toxic in nature, have the tendency to accumulate to such a level in the food chain that they can cause serious health issues, especially in humans [1]. These HMs are released due to natural and anthropogenic activities and are classified as non-degradable inorganic pollutants [2]. Among other HMs, cadmium

(Cd) is considered a non-essential HM for plants that can be extremely toxic for all living organisms [3]. According to an estimate, 25,000 to 30,000 tons of Cd per year are being released into Earth's ecosystem through different sources such as anthropogenic activities, mining operations, the burning of fossil fuels and sewage sludges [4]. About half of of such Cd emissions are due to the weathering of rocks, volcanoes and forest fires [5,6]. In the top 20 contaminants, Cd is ranked 7th among the most hazardous elements by the US-EPA [7]. The widespread release of Cd has caused vast areas to become unproductive or even hazardous for wildlife and humans, thereby causing the disposal of cultivable land to fade away [5]. The use of phosphorus-based fertilizers in the cultivation of plants for energy production has become an important source of Cd accumulation in agricultural soils. Such phosphorous-based fertilizers are primarily manufactured through the chemical treatment of phosphate rocks with a significant concentration of Cd [8].

Phytoremediation is an environmentally friendly technology in which plant roots uptake heavy metals from soils/water into shoot biomass [9,10]. The resulting biomass can be utilized in biorefineries for energy and heat production [11,12]. As the plants absorb nutrients and water from the soil, they have the tendency to uptake and accumulate HMs such as Cd in the roots, which are eventually translocated to the arial parts [13,14]. El. Rasafi et al. [15] reported that excessive Cd absorption affects the plants at molecular, biochemical, physiological and morphological scales. Studies have shown that the excessive absorption of Cd negatively impacts the plant's height, inhibits seed germination, retards root elongation and reduces the leaf count, which may lead to plant death [16,17]. In addition, Cd is known to induce acute and chronic indications in humas, which adversely affects the cardiovascular, musculoskeletal and pulmonary systems as well as kidney function and carcinogenicity [18]. Therefore, there is an urgent need to control the runoff of heavy metals.

Cd not only interrupts plant growth by damaging various leaf structures and reducing chlorophyll contents and leaf gas exchange characteristics [19], but it is also responsible for inducing oxidative stress. The overproduction of reactive oxygen species (ROS) causes damage to the membranes, and further electrotype leakage results in membrane damage that depreciates a plant's defense mechanism, thereby reducing the activities of antioxidants [20]. In general, increases in the antioxidant activities of various enzymes such as superoxide dismutase (SOD) and catalase (CAT) play a vital role in the protection of plants from damage instigated by oxidative stress induced by abiotic factors such as Cd stress [21]. Studies have demonstrated that the relative increase or decrease in the antioxidant enzymes activity may vary between species and the level of HM pollution in soils [12,19,22]. Therefore, in pursuit of identifying tree species that can tolerate HM stress under arid environments, it is important to examine antioxidant activities under HM-contaminated environments. Consequently, in the present experiment, we examined the growth and phytoextraction capability of four tree species (*Conocarpus erectus*, *Syzygium cumini*, *Populus deltoides* and *Morus alba*) in cadmium-contaminated soil. These species are multipurpose, fast growing and widely found in arid to semi-arid environments. Therefore, the findings of the present study are very important for the selection of species suitable for Cd-contaminated soils of arid to semi-arid regions. Recently, fast-growing woody species (*Populus deltoides* and *Morus alba*) are being proposed for phytoremediation purposes and have exhibited tolerance to cadmium (0.018 mg kg$^{-1}$) and lead [23,24]. In this context, the present experiment was conducted to explore the growth; morpho-anatomical, physiological and biochemical responses; and phytoextraction ability of four fast-growing tree species of arid to semi-arid environments in Cd-polluted soil.

## 2. Materials and Methods

### 2.1. Plant Material and Growth Conditions

A pot experiment was carried out in the forest nursery (31°26′ N, 73°06′ E; 184.4 m) at the Department of Forestry & Range Management, University of Agriculture Faisalabad. The average temperature remained between 35 ± 5 °C, with a light period of 15 h. The light

intensity was around 1200 $\mu$mol m$^{-2}$ s$^{-1}$. Six-month-old healthy and undamaged saplings of four forest tree species (*Conocarpus erectus*, (*C. erectus*), *Populus deltoides*, (*P. deltoides*), *Morus alba* (*M. alba*) and *Syzygium cumini* (*S. cumini*)) were selected from the Punjab Forest Research Institute Gatwala Faisalabad and were planted in plastic pots (26 × 40 cm). The pots were filled with sandy loam soil mixed with FYM (farmyard manure) at a ratio of 3:1. The weight of each pot was maintained at 10 kg by adding the soil. The soil mixture used during the experiment was analyzed for nitrogen, phosphorous and organic matter (N, 0.78%; P, 12 ppm; and OM, 8%, respectively). Various other soil properties were also measured such as electrical conductivity (2 dS m$^{-1}$) and pH (6.6). The nutrient supply during the experiment was optimized by adding Smartcot fertilizer in each pot at a rate of 5 g per kg of soil. The pots were watered to field capacity throughout the experiment, and weeding was performed as per the requirement.

### 2.2. Cadmium Application

All saplings were allowed to grow under natural conditions. A total of 80 young saplings were selected for the experiment (20 saplings/treatment). The treatment combinations were as follows: the control—CK, 0 $\mu$M; the cadmium treatment—200 $\mu$M. CdCl$_2$ was used for the application of stress. The molecular weight of CdCl$_2$ = 183.32 g/mol. To prepare the 1M solution of CdCl$_2$, 183.32g of CdCl$_2$ was dissolved in 1 L of distilled water.

### 2.3. Growth and Dry Biomass Production

At the beginning and end of the experiment, the plant height (cm), stem diameter (mm) and number of leaves were measured. At harvest, the saplings were separated into three organs (leaves, stem and roots), and the fresh weight was measured. The saplings were packed into paper bags and dried in a heat oven for 72 h at 80 °C to measure the dry weight of each organ and the total dry weight of each plant was calculated [25]. The root–shoot ratio (R:S) was calculated by dividing the root by the shoot dry weight.

### 2.4. Physiological Parameters

The physiological parameters such as the chl *a*, *b* and carotenoid contents were measured by taking leaf samples of healthy leaves. A total of 0.5 g of the leaf samples were added with 4.5 mL acetone (80%, *v/v*) to make the solution. The resulting solution was centrifuged at 13,000 rpm for 10 min, the supernatant was collected in an Eppendorf tube and the absorbance was measured at 460, 645 and 663 nm, respectively, with the help of a spectrophotometer (Perkin Elmer, Waltham, MA, USA). Chl *a*, *b* was measured by using the formula described by Arnon [26]. The carotenoid contents were determined by following the equation demonstrated by Zafar et al. [27].

### 2.5. Measurement of Cd Concentration in Different Plant Organs (Leaves, Stem and Root)

In the leaves, stem and roots, the concentration of Cd was measured by washing and air-drying the plant samples at 105 °C for 24 h and grinding them using a mortar and pestle. A total of 0.5 g of the powered samples were transferred into 100 mL digestion tubes, and 10 mL of a digestion mixture of 70% HNO$_3$ and HClO$_4$ at a ratio of 2:1 was added. Subsequently, the samples were gradually heated at 170 °C to remove HNO$_3$ (the appearance of white flumes). The tubes were allowed to cool (2 h), and deionized water was added to make up the volume at 100 mL. The solution was filtered using Watman-42 filter paper [28]. The concentration of Cd was measured using the atomic absorption spectrophotometer (Hitachi Polarized Zeeman AAS, Z-8200, Tokyo, Japan), using standard protocols [19].

### 2.6. Measurement of the Translocation Factor

The translocation factor, which is the ability of plants to translocate the HMs from roots towards the shoot, was determined as the ratio of HM content in the shoot (leaves + stem) and root [19].

*2.7. Measurement of Hydrogen Peroxide ($H_2O_2$), Superoxide Radical ($O_2^-$) and Osmolytes Accumulation*

Each 0.2 g of the leaf sample was homogenized with 5 mL of 0.1% Trichloroacetic acid to determine the hydrogen peroxide ($H_2O_2$) following the procedure of Velikova et al. [29]. The absorbance was determined at 390 nm by using the spectrophotometer (Perkin Elmer, Waltham, MA, USA). The concentration of superoxide radicals ($O_2^-$) was determined according to the methodology conducted by Bai et al. [30]. The method of Bates et al. [31] was followed, using ninhydrin to determine the proline content. The total phenolic content was determined in the frozen leaf samples following Ainsworth and Gillespie [32] and using the Folin–Ciocalteu reagent. The total soluble protein was measured according to the method described by Bradford [33]. The soluble sugar was estimated by the anthrone methods according to the protocol of Yemm and Willis [34].

*2.8. Measurement of Antioxidant Enzymes*

The activity of SOD was determined by the reduction rate of nitroblue tetrazolium (NBT), following Beyer and Fridovich [35]. POD activity was measured according to Maehly and Chance [36]. CAT activity was estimated by the method demonstrated by Knörzer et al. [37]. The APX enzyme activity was determined according to [38]. The reduction and absorption at 560 nm, 470 nm and 240 nm were determined using the Eppendorf BioSpectrometer®Basic, Hamburg, Germany.

*2.9. Statistical Analysis*

All the data were tested statistically by applying two-way ANOVA to determine the species (S), treatment (T) and their interaction effects (S × T). The significant differences across the species and the treatments were compared using a multi-comparison test (Tukey's HSD test). Significance was considered at $p < 0.05$, and the means are presented with the SE. All tests were conducted in STATISTICA version 12.5, Statsoft, Maisons Alfort, France.

## 3. Results

*3.1. Cd Treatment and Growth and Biomass Production*

All the traits related to growth and biomass production decreased significantly under the Cd treatment and varied significantly across the four species (Table 1). The mean plant height, stem diameter and number of leaves decreased by 17.6%, 20.9% and 20.2%, respectively, under Cd treatment as compared to CK. Under Cd treatment, the highest decrease in the mean plant height and number of leaves was found in *M. alba* (20% and 36.8%), and the lowest was found in *C. erectus* (13.5% and 6%), whereas the decrease in stem diameter was the highest in *S. cumimi* (32.4%) and the lowest in *C. erectus* (11.9%) as compared to CK. The root–shoot ratio significantly increased in all species under the Cd treatment (Table 1).

The dry biomass production in various plant organs such as the leaf, stem and roots also decreased significantly under the Cd treatment and across the species as compared to CK (Figure 1). Under the Cd treatment, the highest decrease in leaf and stem biomass production was evidenced in *M. alba* (29.2% and 25.8%), and the lowest was evidenced in *C. erectus* (11.6% and 10.4%; Figure 1A,B). In root biomass production, the decrease was the highest in *P. deltoides* (26.5%), whereas a significant increase was evidenced in *C. erectus* (16.6%) under the Cd treatment as compared to CK (Figure 1C). The total biomass production also decreased significantly in all the species under the Cd treatment, with the highest decrease evidenced in *M. alba* and *P. deltoides* (24.2% and 25.7%), whereas the decrease was the lowest in *C. erectus* (5.17%; Figure 1D).

**Table 1.** Means ± SE for growth traits and chl *a, b* and carotenoid contents in four tree species under CK and Cd treatments. Small letters depict significant differences between treatment means (n = 5).

| | | Plant Height (cm) | Stem Diameter (mm) | No. of Leaves | Root–Shoot Ratio | Chl *a* (mg g⁻¹ FW) | Chl *b* (mg g⁻¹ FW) | Carotenoid (mg g⁻¹ FW) |
|---|---|---|---|---|---|---|---|---|
| *C. erectus* | CK | 65.5 ± 1.57 | 5.63 ± 0.17 | 50.7 ± 2.12 | 0.26 ± 0.01 | 1.44 ± 0.07 | 1.65 ± 0.09 | 0.90 ± 0.01 |
| | Cd | 56.6 ± 1.47 | 4.96 ± 0.18 | 47.3 ± 2.09 | 0.38 ± 0.02 | 1.34 ± 0.06 | 1.45 ± 0.02 | 0.76 ± 0.01 |
| *S. cumini* | CK | 50.8 ± 0.73 | 5.76 ± 0.14 | 21.5 ± 0.86 | 0.47 ± 0.01 | 1.73 ± 0.05 | 1.44 ± 0.02 | 0.92 ± 0.04 |
| | Cd | 40.6 ± 1.02 | 3.89 ± 0.18 | 16.7 ± 0.51 | 0.54 ± 0.03 | 1.61 ± 0.13 | 1.31 ± 0.01 | 0.64 ± 0.01 |
| *P. deltoides* | CK | 53.8 ± 1.37 | 4.17 ± 0.15 | 17.0 ± 0.37 | 0.31 ± 0.02 | 1.38 ± 0.01 | 1.36 ± 0.01 | 0.80 ± 0.01 |
| | Cd | 43.6 ± 0.78 | 3.29 ± 0.32 | 12.7 ± 1.12 | 0.47 ± 0.03 | 1.18 ± 0.04 | 1.06 ± 0.09 | 0.46 ± 0.05 |
| *M. alba* | CK | 46.9 ± 2.15 | 3.77 ± 0.11 | 22.5 ± 0.51 | 0.51 ± 0.05 | 1.18 ± 0.01 | 1.12 ± 0.01 | 0.71 ± 0.01 |
| | Cd | 37.5 ± 0.78 | 3.14 ± 0.06 | 14.2 ± 0.71 | 0.63 ± 0.05 | 0.96 ± 0.12 | 0.95 ± 0.05 | 0.46 ± 0.05 |
| S effect | | *p* < 0.001 | *p* < 0.001 | *p* < 0.001 | *p* < 0.001 | *p* < 0.001 | *p* < 0.001 | *p* < 0.001 |
| T effect | | *p* < 0.001 | *p* < 0.001 | *p* < 0.001 | *p* < 0.001 | *p* < 0.001 | *p* < 0.001 | *p* < 0.001 |
| S × T Effect | | *p* = 0.988 | *p* = 0.030 | *p* = 0.529 | *p* = 0.173 | *p* = 0.218 | *p* < 0.001 | *p* = 0.096 |

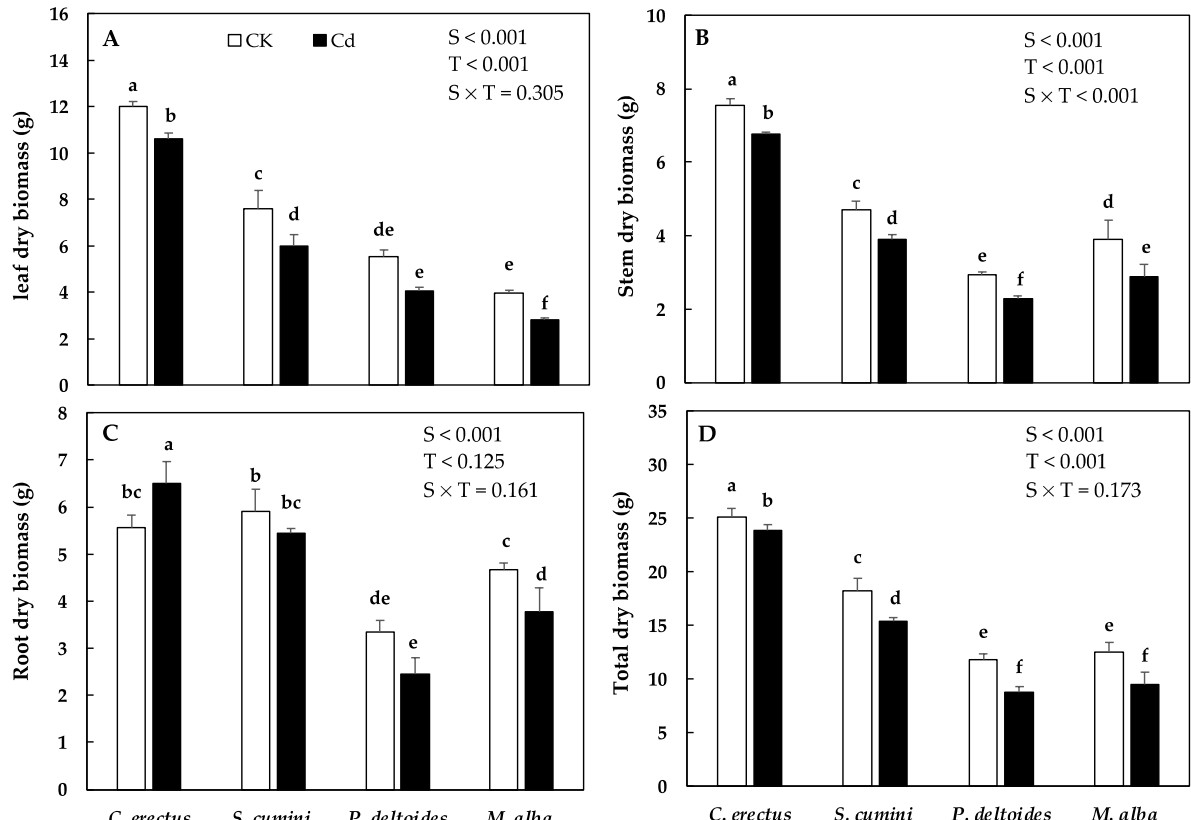

**Figure 1.** Mean values of (**A**) leaf dry biomass, (**B**) stem dry biomass, (**C**) root dry biomass and (**D**) total dry biomass under CK and Cd treatments. Small letters depict significant differences between treatment means (n = 5).

### 3.2. Cd Treatment and Chl a, b and Carotenoid Contents

The physiological pigments such as the chl *a, b* and carotenoid contents significantly decreased under the Cd treatment as well as across the species as compared to CK (Table 1). Under the Cd treatment, the highest decrease in chl *a* was evidenced in *M. alba* (18.6%), whereas for the chl *b* and carotenoid contents, the decrease was found to be the highest in *P. deltoides* (22% and 42.5%, respectively) as compared to CK. Among the four species, the lowest decrease in the chl *a, b* and carotenoids contents was found in *C. erectus* (6.9%, 12.1% and 15.2%, respectively).

### 3.3. Cd Treatment and Proline and Osmolytes Accumulation

The accumulation of proline increased significantly (33.5%) under the Cd treatment as compared to CK. However, the accumulation of proline was found to be the highest in *C. erectus* (47.3%) and was found to be the lowest in *M. alba* (14.6%; Figure 2A). The accumulation of total soluble sugar, total phenolic contents and soluble protein increased significantly by 15.8%, 24.0% and 28.0%, respectively, in plants under the Cd treatment as compared to CK. Among the four species, the increase in soluble protein, total phenolic contents and total soluble sugar was found to be the highest in *C. erectus* (39.1%, 34.9%, and 18.3%, respectively) and the lowest in the *M. alba* saplings (17.8%, 8.47%, and 12.1%, respectively) as compared to CK (Figure 2B–D).

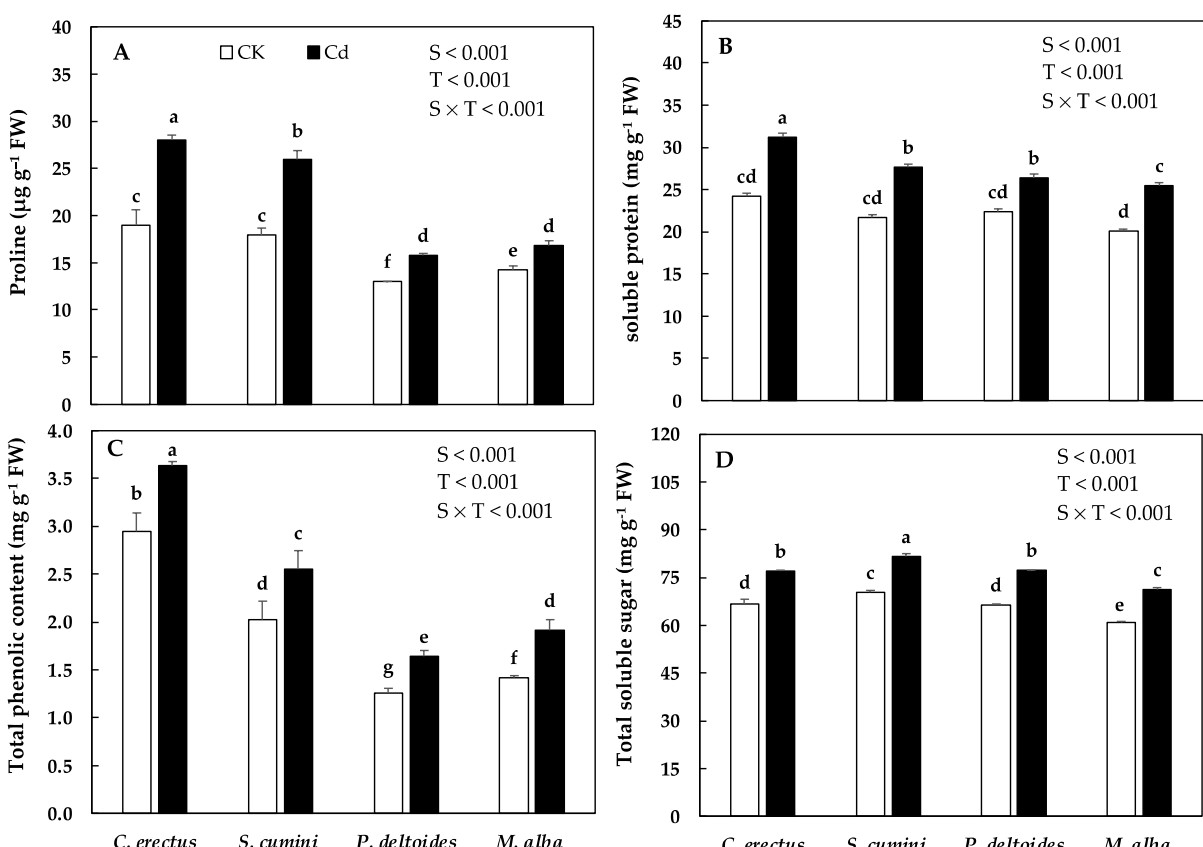

**Figure 2.** Mean values of (**A**) proline, (**B**) soluble protein, (**C**) total phenolic contents and (**D**) total soluble sugar under CK and Cd treatments. Small letters depict significant differences between treatment means (n = 5).

### 3.4. Cd Treatment and Oxidants and Antioxidants Enzyme Activity

The concentration of oxidants such as hydrogen peroxide ($H_2O_2$) and superoxide radical ($O_2^-$) significantly varied under the Cd treatment as well as among the species. Under the Cd treatment, the concentration of $H_2O_2$ and $O_2^-$ increased significantly by 29.1% and 12.6% as compared to CK, respectively. Among the four species, the highest increase in the concentration of $H_2O_2$ and $O_2^-$ was evidenced in *M. alba* (31.1% and 25.1%, respectively), and the lowest was evidenced in the *C. erectus* saplings (26.5% and 5.35%, respectively; Table 2).

The activity of antioxidant enzymes such as SOD, POD, CAT and APX varied and increased significantly in the plants under the Cd treatment as compared to CK (Figure 3). The production of SOD, POD, CAT and APX was found to be the highest in *C. erectus* (42.4%, 33.3%, 33.5% and 39.2%, respectively), and the lowest was found in *M. alba* (23.1%, 13.1%, 4.2% and 24.6%, respectively; Figure 3A–D).

**Table 2.** Means ± SE for oxidants, Cd content in the leaves, stem and root and the translocation factor in four tree species under CK and Cd treatments. Small letters depict significant differences between treatment means (n = 5).

| | | H$_2$O$_2$ (μmol g$^{-1}$ FW) | O$_2^-$ (μmol g$^{-1}$ FW) | Cd Content Leaves (mg Kg$^{-1}$) | Cd Content Stem (mg Kg$^{-1}$) | Cd Content Root (mg Kg$^{-1}$) | Translocation Factor |
|---|---|---|---|---|---|---|---|
| *C. erectus* | CK | 5.62 ± 0.80 | 1.12 ± 0.01 | 0.35 ± 0.01 | 0.79 ± 0.01 | 0.67 ± 0.03 | 1.04 ± 0.11 |
| | Cd | 7.11 ± 0.62 | 1.18 ± 0.01 | 57.1 ± 2.65 | 37.0 ± 2.41 | 75.9 ± 0.05 | 1.24 ± 0.34 |
| *S. cumini* | CK | 7.33 ± 0.15 | 1.16 ± 0.06 | 0.45 ± 0.02 | 0.93 ± 0.01 | 0.44 ± 0.02 | 0.97 ± 0.16 |
| | Cd | 9.94 ± 0.34 | 1.28 ± 0.02 | 42.6 ± 0.67 | 20.3 ± 0.49 | 60.1 ± 0.47 | 1.03 ± 0.21 |
| *P. deltoides* | CK | 8.95 ± 0.16 | 1.22 ± 0.02 | 0.55 ± 0.01 | 0.93 ± 0.01 | 0.88 ± 0.01 | 1.53 ± 0.09 |
| | Cd | 11.1 ± 0.61 | 1.33 ± 0.01 | 44.5 ± 0.25 | 24.3 ± 1.57 | 67.1 ± 1.03 | 1.02 ± 0.13 |
| *M. alba* | CK | 8.95 ± 0.04 | 1.27 ± 0.01 | 0.55 ± 0.01 | 0.88 ± 0.01 | 0.98 ± 0.01 | 0.73 ± 0.06 |
| | Cd | 11.0 ± 0.09 | 1.59 ± 0.04 | 39.1 ± 1.93 | 22.2 ± 1.78 | 62.2 ± 2.49 | 0.98 ± 0.11 |
| S effect | | $p < 0.001$ | $p < 0.001$ | $p < 0.001$ | $p < 0.001$ | $p < 0.001$ | $p < 0.001$ |
| T effect | | $p < 0.001$ | $p < 0.001$ | $p < 0.001$ | $p < 0.001$ | $p < 0.001$ | $p = 0.977$ |
| S × T Effect | | $p = 0.134$ | $p < 0.001$ | $p < 0.001$ | $p < 0.001$ | $p < 0.001$ | $p < 0.001$ |

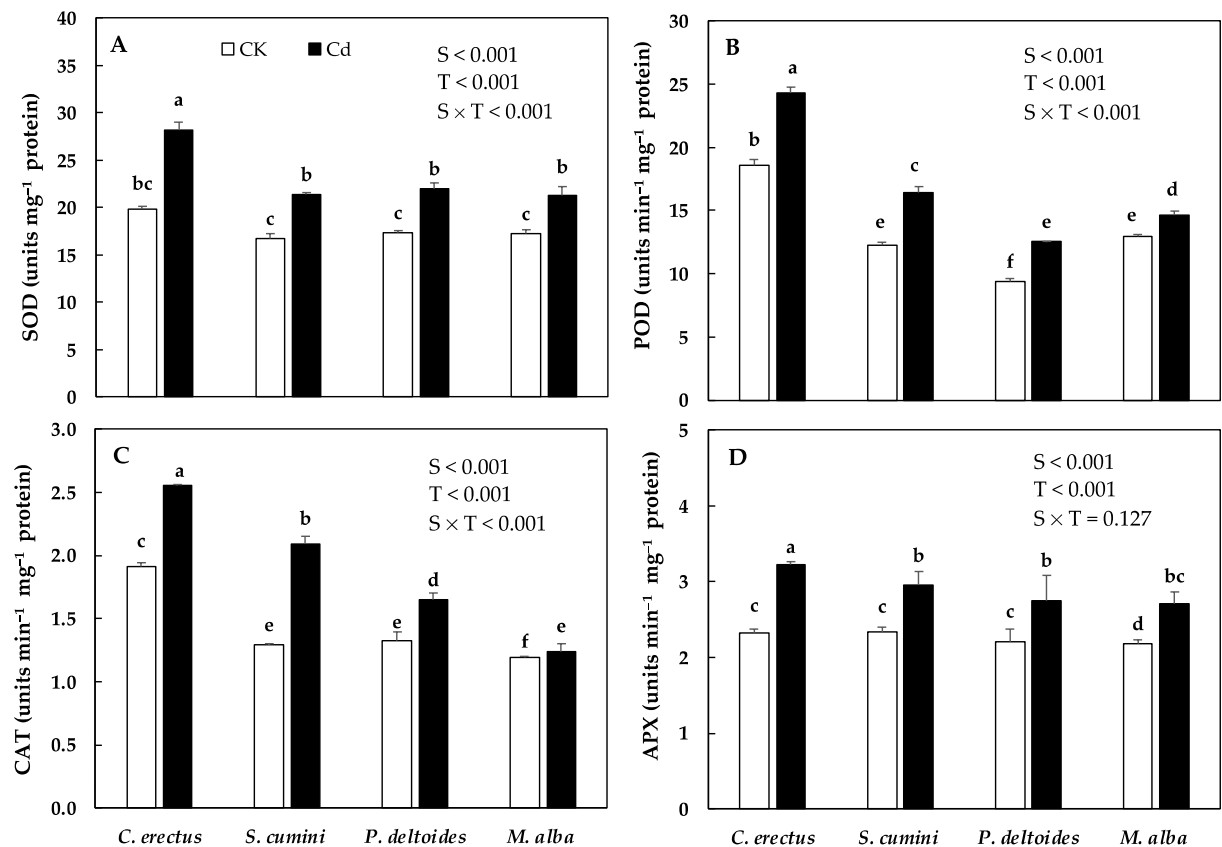

**Figure 3.** Mean values of (**A**) superoxide dismutase, SOD; (**B**) peroxidase, POD; (**C**) catalase, CAT; and (**D**) ascorbate peroxidase, APX under CK and Cd treatments. Small letters depict significant differences between treatment means (n = 5).

### 3.5. Content of Cd in Various Plant Organs and the Translocation Factor

The content of Cd in the leaves, stem and roots increased significantly under the Cd treatment as well as across the species (Table 2). Under the Cd treatment, the mean content of Cd in the leaves, stem and roots was found to be the highest in *C. erectus*, followed by *S. cumini* and *P. deltoides*. The mean content of Cd in the leaves, stem and roots remained the lowest in *M. alba* (Table 2). The translocation factor (ratio of Cd content in the shoot and

root) also varied significantly between species and across the treatments. Among the four species, the translocation factor was higher than 1 in *C. erectus*, *S. cumini* and *P. deltoides*, which shows that the translocation of Cd from roots towards leaves was higher in these three tree species compared to *M. alba*, where the translocation factor remained less than 1.

## 4. Discussion

Studies have demonstrated that plant growth and biomass production are key traits for evaluating a species' potential as a phytoextractor or phytoremediator [39]. Recent studies have shown that a high content of heavy metals in the soil can have a significant harmful impact on plant growth and productivity [40]. In this study, we observed that plant growth and dry biomass production (leaf, stem, root and total biomass) decreased significantly under Cd treatment. However, the highest decrease was evidenced in *M. alba* and *P. deltoides* as compared to the *C. erectus* and *S. cumini* saplings, respectively. Similar negative effects of Cd have been reported previously in *P. deltoides* × *P. nigra* [41], willow clones [42], *P. nigra* and *M. alba* [23,24]. Growth parameters are often used as indicators for determining a species' tolerance against abiotic stresses. [43–45]. Previous studies have linked the heavy metal toxicity with the degradation of chromatin, which may result in a reduction in cell division and a loss of plant productivity [46]. Furthermore, reports of damage to the photosynthesis apparatus and changes in other metabolic activities have also been demonstrated under Cd treatment [47]. In non-hyper-accumulating species, water and nutrient imbalance, increased oxidative stress and enzymatic activities have been commonly observed in response to Cd treatment [41,44].

Similarly, chl *a, b* and carotenoid content play a significant role in enhancing a plant's tolerance to heavy metal stress. In this study, we observed a significant decrease in the chl *a, b* and carotenoid contents in all four tree species. Our results are in accordance with the findings where chloroplast damage due to Cd treatment resulted in a significant decrease in photosynthetic pigments such as chl *a, b* and carotenoid contents [48]. However, in this study, the reduction in the chl *a, b* and carotenoid contents was higher in the case of the *M. alba* and *P. deltoides* saplings as compared to *C. erectus* and *S. cumini*. In the previous studies on *Populus nigra*, *Salix alba*, *Eucalyptus* hybrid, *Conocarpus erectus*, *Eucalyptus camaldulensis* and *Conocarpus lancifolius*, similar decreases have been reported, where the chl *a, b* and carotenoid contents decreased in plants grown in Cd-contaminated soil [44,49–52]. Therefore, based on the lowest reduction in the chl *a, b* and carotenoid contents observed in *C. erectus*, it can be assumed that *C. erectus* showed a higher tolerance against the given Cd treatment.

The production of antioxidant enzymes in response to the increased production of oxidants plays a significant role in the adaptation and survival of species under biotic and abiotic stress. Under the Cd treatment, the production of reactive oxygen species (ROS) increased as the production of the concentration of oxidants such as $H_2O_2$ and $O_2^-$ increased in all four tree species. However, the observed overproduction of $H_2O_2$ and $O_2^-$ was the highest in *M. alba* and was the lowest in the *C. erectus* saplings. These results indicate that *C. erectus* is better adapted to low levels of Cd treatment. The significant increase in the production of ROS under Cd treatment observed in this study is in accordance with the previous studies on *Populus* × *canescens* [53]. Such increase in the production of ROS has been linked to the impairment of the mitochondria and the photosynthetic electron transfer chain due to Cd stress [53–55]. To avoid ROS-induced injury due to the overproduction of $H_2O_2$ and $O_2^-$, plants tend to elevate the production of various types of antioxidant enzymes that help prevent cellular damage under abiotic stress [56]. Studies have demonstrated that the most important antioxidant enzyme in scavenging the overproduction of oxidants is SOD, followed by POD, CAT and APX [57]. The SOD scavenges the production of $O_2^-$ into $H_2O_2$, which is further neutralized to $H_2O$ by the other antioxidant enzymes such as CAT and POD [58]. CAT is generally present in the single-membrane-bounded structure (peroxisome) and is responsible for scavenging the increased concentration of $H_2O_2$ by converting $H_2O_2$ to $H_2O$ in the plant cell [52,59]. The

activity of all of the antioxidant enzymes (SOD, POD, CAT and APX) increased significantly in all four tree species under the Cd treatment. However, the activity of antioxidant enzymes remained the highest in *C. erectus* as compared to the other three species. These results are in accordance with the previous study in which antioxidant enzyme activity increased in poplar and willow clones and *Acacia nilotica* and *Eucalyptus* hybrids under Cd stress [41,42,45,48]. In the present study, the increased production of antioxidant enzymes in response to the production of oxidants shows a better stress tolerance strategy in *C. erectus* saplings as compared to *M. alba* and *P. deltoides*.

Several compatible solutes such as proline, soluble sugar, total phenolic contents and soluble protein are regarded as important nonenzymatic antioxidant enzymes that help in protecting the subcellular structure and scavenge the ROS in the plant cell under abiotic and biotic stress conditions [52]. In the present study, the concentration of proline, soluble sugar, total phenolic contents and soluble protein increased significantly in all four tree species under the Cd treatment, with the highest increase evidenced in the *C. erectus* and *S. cumini* saplings. In the previous studies, such increase in proline and soluble sugar content has been related to the decrease in osmotic potential, which enhances the physiological functioning and growth traits and regulates the water absorption by plants during abiotic stress [58]. On the other hand, the phenolic compounds are considered as secondary metabolites which are important in enhancing the plant tolerance under abiotic stress [55]. Different studies have reported up-regulation in the expression of enzymes synthesizing phenolic compounds (phenylalanine ammonia-lyase), as they act as antioxidant enzymes under stressful environments [60]. Our results are in accordance with the previous studies on *P. × canescens* and *C. camphora* species, in which the accumulation of soluble protein and soluble sugar was increased under Cd treatment [52,61,62].

The abilities of species to accumulate and translocate heavy metals have been discussed in studies that were focused on selecting species for phytoextraction in arid to semi-arid environments [51]. In this study, although the Cd content in leaves was found to be the highest in *C. erectus*, followed by the *P. deltoides*, *S. cumini* and *M. alba* saplings, the accumulation of Cd was higher in the roots than that in the stem and leaves in both *C. erectus* and *P. deltoides* as compared to *S. cumini* and *M. alba*. Similar results have been demonstrated in eucalyptus species, where the accumulation of Cd in the roots was higher as compared to that in the leaves and stem [44,61,62]. Furthermore, Manousaki et al. [63] also evidenced a higher content of Cd in the root of *Atriplex halimus*. The mechanisms of Cd collection and immobilization in roots have been related to (i) the chelation of Cd by thiol-containing peptides (metallothioneins and phytochelatins), (ii) Cd assortment in root cell vacuoles and (iii) adsorption in cell walls [16,64,65]. Contrary to our results, the content of Cd was found to be the highest in the leaves of two Salix species and *N. tabacum* plants [16,17]. Moreover, in this study, the Cd content in the stem was found to be the highest in *C. erectus*, followed by *P. deltoides*, *M. alba* and *S. cumini*, respectively. The results of our experiment suggest that all species adopted an exclusion tactic by retaining the highest content of Cd in the roots and translocating the lower content of Cd towards the stem and leaves. The translocation factor (TF) of heavy metal is another important component for determining the potential of the phytoremediation of heavy metal, as the TF depends upon on the relative accumulation of heavy metal in roots and shoots [17,51,52]. If a species has a translocation factor >1, it shows a high mobility of heavy metal from the roots towards the shoots. In this study, the translocation factors of *C. erectus*, *S. cumini* and *P. deltoides* were higher than 1, which means that they are suitable species for the phytoextraction of Cd.

## 5. Conclusions

Cd treatment had a significant negative effect on all four species, whether it was in terms of growth, dry biomass production (leaf, stem and root) or chlorophyll contents. However, the lowest decrease was observed in *C. erectus*. Furthermore, the concentration of osmolytes (proline, soluble sugar, total phenolic contents and soluble protein) and

the activity of antioxidant enzymes increased significantly, whereas the concentration of oxidants such as hydrogen peroxide and superoxide radical decreased significantly in all four forest species under the Cd treatment. The activity of antioxidant enzymes was found to be the highest in the *C. erectus* and *S. cumini* saplings, depicting an effective detoxification ability. The Cd content was the highest in the roots, followed by the leaves and the stem, whereas a Cd translocation factor of less than 1 in the *M. alba* saplings indicated a low mobility of Cd from the roots towards the shoot. Overall, we may conclude that *C. erectus* is the most suitable species for the phytoaccumulation and rehabilitation of Cd-contaminated sites due to the increased ability of scavenging oxidative stress, the increased production of osmolytes, which may help in maintaining the plant's water potential, and the higher accumulation and translocation of Cd compared to *S. cumini*, *P. deltoides* and *M. alba*.

**Author Contributions:** Conceptualization, F.R. and M.M.M.; methodology, F.R. and Z.Z.; software, Z.R. (Zohaib Raza), M.M. and W.R.K.; validation, M.M. and Z.Z.; formal analysis, M.Z.R. and Z.R. (Zamri Rosli); investigation, M.M. and F.R.; resources, Z.Z. and F.R.; data curation, F.R., Z.Z., S.A., F.A.B. and M.M.; writing—original draft preparation, Z.Z. and M.M.; writing—review and editing, F.R., S.A., Z.R. (Zohaib Raza) and M.M.; visualization, F.R., Z.R. (Zamri Rosli) and W.R.K.; supervision, F.R.; project administration, Z.Z. and M.M.M.; funding acquisition, Z.Z., F.R. and M.M.M. All authors have read and agreed to the published version of the manuscript.

**Funding:** This research received funding from the Offices of Research, Innovation and Commercialization (ORIC) Grant no. A/C AO5213, University of Agriculture Faisalabad, Pakistan and Universiti Putra Malaysia, Bintulu 97008, Malaysia, Vote No. 6700205.

**Acknowledgments:** We are thankful to M. Siddique for providing technical support during the experiment, to Rana Muhammad Atif for providing laboratory facilities for the antioxidant enzymes and the concentration of heavy metal (Cd) and Universiti Putra Malaysia for supporting this study.

**Conflicts of Interest:** The authors declare no conflict of interest.

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
