# Peer review of "The Change in Growth, Osmolyte Production and Antioxidant Enzymes Activity Explains the Cadmium Tolerance in Four Tree Species at the Saplings Stage"

_forests, doi:10.3390/f13091343_

Round 1

Reviewer 1 Report

Except for the Latin spelling errors of the plant species, there is no deficiency in the article. It is shown at attachement.

Populus × canadensis, known as Canadian poplar or Carolina poplar,  is a naturally occurring hybrid of Populus nigra and Populus deltoides.nThis is must be explain in the text.

 Populus × canescens, the grey poplar, is a hybrid between Populus alba (white poplar) and P. tremula (common aspen). It is not compare with P. x canadensis !

Author Response

Response: Thank you for appreciating our work. The suggestions have been incorporated in the revised version of the manuscript.

Reviewer 2 Report

Dear Corresponding Author

Your paper is about phytoremediation and it is well-developed manuscript but please check again and correct some spelling errors such as italic names and spaces between 200mM, for instance.

Regards

Author Response

Response: Thank you for appreciating our work. The inconsistencies have been removed in the revised version of the manuscript.

Reviewer 3 Report

The manuscript entitled “The nexus between growth, osmolyte production and antioxidant enzymes activity explains the cadmium tolerance in four tree species at saplings stage” presents data on Cd uptake, ROS accumulation, contents of total amount of major metabolites and the activity of antioxidant enzymes upon treatment of young tree seedlings with cadmium chloride. The paper aims at indicating the woody species with a high potential for utilization as a phytoextractor of Cd showing the highest tolerance among the four species analyzed. The results are interesting, however the manuscript have several major drawbacks and should be greatly adjusted before reconsideration for publication in the Forests journal. Major issues were listed as follows:

11. Title: is an overestimation of the results since “the nexus” was not evaluated with statistical methods, e.g. PCA. Additional analysis is needed or change the title to more appropriate to the paper contents

22.  Abstract: provide the age of plants in abstract and type of inorganic salt used for treatment

33. Introduction: describe the species of interest, describe the level of soil pollution with Cd and plant tolerance levels

44.  Methods: provide details for chemical analyses, suppliers and names of instrumentation

55. Results: Add PCA or other analysis to evaluate the relations between investigated traits

66. Discuss results with woody species

77. Discussion repeats partly the results section, can be combined? Do not address figures and tables in discussion section

88. Lack of Conclusion section

Whole text:

19. The language should be checked because of plenty of errors that lower the value of the manuscript can be found

210. Change “contamination” to the word “pollution”

111. Change “better tolerance” to “higher”

412. Avoid repeating the parts of sentences

513.  Distinguish metal concentration (for solutions), content (for tissue accumulation or level in soil)

614. Do not use capital c in the word cadmium

715. Plants do not absorb heavy metals rather uptake

816. The overproduction of ROS caused damage of membranes and further electrolyte leakage

917. MDA is not an antioxidant

118. Use term “Cd treatment” instead of “Cd stress” throughout the manuscript since stress  is the effect of treatment and is different for the investigated species

119. Use simple past and simple present tenses where possible

120.   2.1. Plant material and growth conditions

121.   Indicate dimeter and height of pots

122.   2.2 line 118-121 must be removed it’s basic chemical knowledge

123.   Change plant sections to organs

124.   How was the tissue homogenized or just “mixed” as written? Was it frozen when terminated in liquid nitrogen?

125.   The authors measured the activity of antioxidant enzymes rather than concentration (this error occurs multiple times)

126.   Move TF calculation from 2.8 to 2.5 subsection

127.   Was TF calculated as shoot-to-root or leaf-to-root ratio? Was it average? If so the biomass of organs was taken into calculations? Provide formula in text

228.   In results subsections remove “Cd stress” form titles or replace with “treatment”

229.   Why authors performed post-hoc analysis for SxT effect if it was not significant in case of about half of parameters?

230.   Put “n” value in figure captions and table legends

231.   Remove italic for Fig 2 caption

232.   Use figures for all investigated traits instead of tables

233.   Divide biomass parameters from pigment contents into two figures, the same with ROS and Cd accumulation

234.   Keep the order of figures accordingly to the text contents

Author Response

We are highly grateful for your time and suggestions to improve our manuscript. All the suggestions have been incorporated in the revised manuscript. Point wise response to the quarries and suggestions are listed below.

  1. Title: is an overestimation of the results since “the nexus” was not evaluated with statistical methods, e.g. PCA. Additional analysis is needed or change the title to more appropriate to the paper contents

Reponses: Thank you for pointing out. The addition of PCA can be a valuable. However, at this stage, we opted to change the title of the paper. Your suggestion will be taken in account in the future.

  1. Abstract: provide the age of plants in abstract and type of inorganic salt used for treatment

Response: The desired information has been added in the revised version.

  1. Introduction: describe the species of interest, describe the level of soil pollution with Cd and plant tolerance levels

Response: The species have been briefly introduced and the level of Cd pollution has also been added the revised version.

  1. Methods: provide details for chemical analyses, suppliers and names of instrumentation

Response: The desired information has been added in the revised version.

  1. Discuss results with woody species

Response: The suggestion has been incorporated in the revised version. 

  1. Discussion repeats partly the results section, can be combined? Do not address figures and tables in discussion section

Response: In the discussion section, the results were recalled briefly and were related to the previous studies. The reference of the tables and figures have been removed from the discussion section.

  1. Lack of Conclusion section

Response: The section has been added although it is not mandatory according to the journal guidelines.

  1. The language should be checked because of plenty of errors that lower the value of the manuscript can be found

Response: The language has been corrected as advised.

  1. Change “contamination” to the word “pollution”

Response: Corrected as advised.

  1. Change “better tolerance” to “higher”

Response: Corrected as advised.

  1. Avoid repeating the parts of sentences

Response: Corrected as advised.

  1. Distinguish metal concentration (for solutions), content (for tissue accumulation or level in soil)

Response: Corrected as advised.

  1. Do not use capital c in the word cadmium

Response: Corrected as advised.

  1. Plants do not absorb heavy metals rather uptake

Response: Corrected as advised.

  1. The overproduction of ROS caused damage of membranes and further electrolyte leakage

Response: Corrected as advised.

  1. MDA is not an antioxidant

Response: Corrected as advised.

  1. Use term “Cd treatment” instead of “Cd stress” throughout the manuscript since stress is the effect of treatment and is different for the investigated species

Response: Corrected as advised.

  1. 2.1. Plant material and growth conditions

Response: Corrected as advised.

  1. Indicate dimeter and height of pots

Response: The desired information has been added in the revised version.

  1. 2.2 line 118-121 must be removed it’s basic chemical knowledge

Response: Corrected as advised.

  1. Change plant sections to organs

Response: Corrected as advised.

  1. How was the tissue homogenized or just “mixed” as written? Was it frozen when terminated in liquid nitrogen?

Response: Samples were homogenized in the ice bath. The discrepancy has been corrected.

  1. The authors measured the activity of antioxidant enzymes rather than concentration (this error occurs multiple times)

Response: Corrected as advised.

  1. Move TF calculation from 2.8 to 2.5 subsection

Response: Corrected as advised.

  Was TF calculated as shoot-to-root or leaf-to-root ratio? Was it average? If so the biomass of organs was taken into calculations? Provide formula in text

Response: Concentration of Cd in leaves, stem and roots were determined and TF was calculated as the ratio between Cd concentration in leaves+stem and roots. The total biomass of each organ was not taken into the calculation.

  1. In results subsections remove “Cd stress” form titles or replace with “treatment”

Response: Corrected as advised.

  1. Why authors performed post-hoc analysis for S xT effect if it was not significant in case of about half of parameters?

Response: The interaction effect was a part of the model where two factors were taken as fixed effect. Post-hoc was done to evidence the trend between each species.

  1. Put “n” value in figure captions and table legends

Response: The desired information has been added in the revised version.

  1. Remove italic for Fig 2 caption

Response: Corrected as advised.

  1. Use figures for all investigated traits instead of tables

Response: If we make figures for all the measured traits, the number of figures will increase way too much. Therefore, a mix of tables and figures were opted.

  1. Keep the order of figures accordingly to the text contents

Response: Corrected as advised.

Round 2

Reviewer 3 Report

The manuscript was partly adjusted to comments.

The text should be carrfully chacked for editing as well as language errors.

Still a few important issues needs correction:

 1. distingush the content (in tissue, in soil) and concentration (in solution). I think authors may confuse the concentration which is measured in solution in AAS with the content of metal in tissue which is finally caluculated as a result

2. Why the sum of metal content in leaves and shoots is used fot TF calculation? Please explain or give literature referance. Why not to use mean value? 

3. Do not mix physiological data with cadmium accumulation in one table.

4. Do not perform post-hoc test for not significant effect according to anova. Just show mean values and SE in chart/table. It's a serious methodological error. Needs correction.

Author Response

Thank you once again for your comments and suggestions for the improvement of our manuscript.

The manuscript was partly adjusted to comments.

The text should be carefully checked for editing as well as language errors.

Response: The text was checked for language errors.

Still a few important issues need correction:

  1. distinguish the content (in tissue, in soil) and concentration (in solution). I think authors may confuse the concentration which is measured in solution in AAS with the content of metal in tissue which is finally calculated as a result

Response: The error has been corrected throughout the manuscript as indicated.  

  1. Why the sum of metal content in leaves and shoots is used fot TF calculation? Please explain or give literature reference. Why not to use mean value? 

Response: The reference of the method has been added in the revised version.

  1. Do not mix physiological data with cadmium accumulation in one table.

Response: Technically, the suggestion is correct but to reduce the number of tables and figures we had to present both datas in a single table.

  1. Do not perform post-hoc test for not significant effect according to anova. Just show mean values and SE in chart/table. It's a serious methodological error. Needs correction.

Response: The correction has been made in both tables as suggested.